# Tennessee Pharmacists’ Opinions on Barriers and Facilitators to Initiate PrEP: A Qualitative Study

**DOI:** 10.3390/ijerph19148431

**Published:** 2022-07-10

**Authors:** Alina Cernasev, Crystal Walker, Caylin Kerr, Rachel E. Barenie, Drew Armstrong, Jay Golden

**Affiliations:** 1Department of Clinical Pharmacy and Translational Science, College of Pharmacy, University of Tennessee Health Science Center, Nashville, TN 37211, USA; 2College of Nursing, University of Tennessee Health Science Center, Nashville, TN 37211, USA; cmarti47@uthsc.edu; 3Kroger Pharmacy, 9225 Kingston Pike, Knoxville, TN 37922, USA; caylin.kerr@stores.kroger.com; 4College of Pharmacy, University of Tennessee Health Science Center, Nashville, TN 37211, USA; rbarenie@uthsc.edu (R.E.B.); darmst11@uthsc.edu (D.A.); 5Walgreens Specialty Pharmacy, Nashville, TN 37203, USA; jon.golden@walgreens.com

**Keywords:** PrEP, HIV negative, pharmacist, US

## Abstract

Pre-exposure prophylaxis (PrEP) is recommended to prevent the transmission of the human immunodeficiency virus (HIV). Although an effective treatment, the uptake in the United States remains low. Pharmacists are well-positioned to initiate the conversation with patients about PrEP, but few studies exist exploring their unique roles. The objective of this study was to characterize Tennessee pharmacists’ perceptions about access to PrEP. A qualitative study was used to gather the data that consisted of virtual Focus Groups over four months in 2021 from practicing Tennessee pharmacists. Emails were sent to all Tennessee licensed pharmacists to recruit them to participate in the study. Recruitment continued until Thematic Saturation was obtained. The corpus of data was audio-recorded, transcribed, and analyzed by the research team. Thematic Analysis revealed two themes: (1) Barriers to accessing PrEP; (2) Potential solutions to address barriers identified. These findings highlighted barriers and identified solutions to improve access to PrEP in Tennessee; additional financial assistance programs and marketing programs targeting patients and providers are needed to enhance PrEP access.

## 1. Background

Pre-exposure prophylaxis (PrEP) is a medication that people at risk for human immunodeficiency virus (HIV) take to significantly lower their chances of contracting HIV. The Centers for Disease Control and Prevention (CDC) state that PrEP is highly effective when taken as prescribed as it reduces the risk of contracting HIV from sex by approximately 99% and from injection drug use by at least 74% [1]. PrEP as HIV prevention can aid in ending the HIV epidemic as it reduces the number of new infections; however, most individuals who may benefit from PrEP do not receive it [2]. There are several factors that have contributed to poor PrEP uptake, such as lack of PrEP awareness among patients, negative perception of prophylaxis, misunderstanding of how PrEP medications work, and poor patient–provider relationships [3]. 

Furthermore, of the one million Americans who could benefit from PrEP, only 25% are using this prevention method and in the South, an area geographically burdened with new HIV infections, PrEP access and uptake are much lower [1,4]. In Tennessee, the PrEP-to-need ratio, the number of PrEP users to the number of people newly diagnosed with HIV, is low for certain subgroups such as women and people ages 13–24 [2]. Black and Hispanic/Latino people are at the highest risk of HIV exposure, but these groups have the lowest rates of PrEP use among all racial/ethnic groups: only 9% of approximately 469,000 Black people who could benefit from PrEP received a prescription in 2020, and only 16% of approximately 313,000 Hispanic/Latino people who could benefit from PrEP received a prescription [4]

To improve access to PrEP, especially in the South, additional innovative strategies are necessary. Pharmacists are well-positioned to support these efforts due to their accessibility, expertise in medication management, and integration into various care models, including in community settings, ambulatory care, inpatient settings, and more [5]. Additionally, pharmacists may also address and overcome the phenomenon of the ‘purview paradox’, which is when primary care providers, who are often in the best position to prescribe PrEP, believe that PrEP is beyond their purview [6]. HIV specialists are often caring for people already living with HIV and often do not have an eligible patient population that would benefit from PrEP. This creates a gap in care that could be closed by pharmacists through education of patients and providers alike.

The emergence of pharmacists’ roles in enhancing PrEP uptake and knowledge about how to address barriers to treatment may inform the design and implementation of robust programs in the healthcare field to support patients. To date, no published qualitative studies of barriers and facilitators to PrEP access have been conducted with Tennessee pharmacists. Thus, this study aimed to address this gap in the literature by exploring Tennessee pharmacists’ perceptions about PrEP.

## 2. Materials and Methods

This pilot study used a qualitative approach to describe the prominent barriers and facilitators to accessing PrEP from the pharmacist perspective. The Theoretical Domains Framework (TDF) was used in this study. This framework “provide[d] a theoretical lens through which to view the cognitive, affective, social, and environmental influences on behavior” [7]. When utilizing TDF, the first step is to select a target behavior [7]. Pharmacists in this study were asked to identify factors that could influence the behavior of accessing and initiating PrEP and to provide recommendations on how to improve access. The next step is to identify an appropriate study design. Since TDF has been “largely used during exploratory and formative stages of a research program to inform problem analysis and intervention development using qualitative interviews”, TDF is the most appropriate framework for this study [7]. As obstacles to PrEP access from the pharmacists’ perspective are being used to develop a pharmacy-led PrEP program in the state of Tennessee, TDF supports gathering these data to inform the design and implementation of a future intervention [7].

The study was approved by the Institutional Review Boards at the University of Tennessee Health Science Center (# 21-08044-XM, 17 March 2021). Data were collected between April and August 2021, and four Focus Groups (FG) were conducted [8,9]. FG were selected for this study because of their capacity to provide rich data and the researchers’ interest in listening to group discussions of barriers and facilitators to PrEP access through the lens of a pharmacist [10,11,12].

### Subjects, Data Collection, and Analysis

The Tennessee Board of Pharmacy provided a list of licensed pharmacists in the state, including their names and email addresses. We sent an email to all licensed pharmacists to invite them to participate in the study. Of those, only licensed pharmacists practicing in a hospital or community setting in Tennessee and willing to share their perspectives about PrEP were eligible to participate. Once a group of six subjects were interested in participating, a Zoom link was provided where the subjects were able to participate virtually. At the beginning of each virtual FG, oral informed consent was obtained, with each subject agreeing to participate. Four subjects declined to participate after listening to the informed consent [10,11]. All the subjects who participated received an Amazon gift card.

All FG were conducted by two researchers (AC, CW), and one of the researchers took field notes during the discussions that provided additional insights to best interpret the data [11,12]. The FG guide was used to keep the discussion focused on several topics: (a) barriers and facilitators to access PrEP, and (b) practicing pharmacists’ suggestions for enhancing access to PrEP directly from the pharmacy. For example, some of the questions posed to the pharmacists included: (1) What are some challenges about patients at risk for HIV in terms of getting and taking medications? (2) How would you attract a patient to initiate PrEP? (3) Have you been approached by patients asking about PrEP? (4) What are other barriers in your opinion that might prevent you from engaging with patients at high risk for HIV in educating them on HIV PreP? The FG guide was initially reviewed by a panel of nurses, pharmacists, and physicians working in the infectious disease area. The FG guide was modified slightly after each focus group due to the discussions that posed additional questions to explore other emerging issues [10,11]. All FG were recorded and transcribed verbatim by a professional transcription company and imported into Dedoose^®^ (Manhattan, CA, USA), a qualitative software [11,12].

Braun and Clarke’s reflexive thematic analysis was used to analyze the FG [13]. The research team followed the six steps recommended by Braun and Clarke [13]. For example, in the first step of the analysis, the researchers read the transcript to immerse themselves in the data [13]. In the second step, two researchers (AC, CK) coded inductively by identifying the words and/or phrases and assigning a code [13]. During this step, the initial codes were generated and a third researcher (CW) discussed the proposed codes. The team (AC, CK, and CW) ensured a unified system of coding was obtained by discussing each code and reaching a consensus on the interpretation among researchers [13]. Categories were developed to organize codes into meaningful clusters. In the last stage, all the FG were interpreted to generate themes and subthemes regarding barriers and facilitators to accessing PrEP [13]. To ensure rigor of the data analysis, the researchers used the guide provided by Weinberger et al. [14]. For example, the transcription was conducted by an independent company that minimized bias [14].

## 3. Results

A total of 21 subjects participated in four focus groups from April to August 2021. The majority of subjects were practicing in the community pharmacy setting (n = 13) with an average of 10 years of experience, which ranged from 1 to 45 years. A few subjects were practicing in a hospital or clinic (n = 5).

Thematic analysis revealed two major themes that describe the barriers and facilitators pharmacists perceive towards PrEP. The first theme illustrated the barriers to PrEP access along with three sub-themes, including (1) the high cost that PrEP places on patients, (2) the stigma associated with PrEP, and (3) lack of patient knowledge about PrEP. The second theme addressed the pharmacists’ perceptions about solutions to address the barriers previously identified along with two sub-themes, including (1) addressing financial constraints and (2) lack of knowledge.

### 3.1. Theme 1: Barriers to Accessing PrEP

This theme revealed pharmacist-identified factors that might influence whether or not patients initiate PrEP. For example, one subject discussed the importance of intrinsic factors that may motivate patients to access and initiate PrEP. The subject said:


*“I think motivation is another big thing. I think people are just, often times, not necessarily motivated to go to the doctor because they don’t know what kind of experience they’re going to have. They don’t know what kind of judgement, if any, they’re going to have.”*
(S1, FG1)

Similarly, another subject presented her opinion about personal factors that might hinder PrEP initiation among patients. This quote highlighted the importance of the pharmacist gaining knowledge about the patients’ condition and their individual challenges to better support them.


*“… Another [reason] could be mental health factors, no motivation, or there’s depression. That’s also associated with HIV, but they could be depressed. Taking medication, it could be not remembering. Issues with remembering to take the medication at certain times of the day...”*
(S14, FG3)

A few subjects acknowledged the importance of referring a patient to a healthcare provider even if the patient did not specifically ask to be referred. For example, “*being able to know a specific provider… so… you could refer them*” (P4, FG1) provides an example of proactive care coordination that the pharmacist can play an important role in for greater PrEP access.


*“…there were typically specific providers that prescribed PrEP, so knowing like, if a patient called and kind of had those apprehensive moments about like, oh, well, I don’t know who to go see for this… Like being able to know if specific providers that you typically get those prescriptions from so you can refer them.”*
(S4, FG1)

In the same FG discussion, another subject expressed the practical value of referring a patient to a provider while avoiding the stigma associated with PrEP. This extract demonstrates the importance of being aware of the practice site and how to minimize the shame associated with PrEP, which could also serve as a motivator for PrEP access.


*“…to kind of piggyback off that, what I was thinking as well, in a small town like that, a small environment where everybody knows everybody, you almost have to do it remotely in order to guarantee privacy. So you want to make sure that nobody in your town, because everybody talks, everybody is going to know everything, even if you just show up to a clinic, or you go to a lab and get lab work, that could be a deterrent for patients, knowing that, okay, which- who’s- not even who is working at the clinic, but who is going to see me going to the clinic, and what could they say? I think you have to do it remotely.”*
(S2, FG1)

Sub-theme 1: Financial constraints to accessing PrEP

One participant emphasized the cost of the medication as a limiting factor to patients accessing PrEP.


*“… just knowing that the cost of these drugs are very expensive might be a limiting factor for some of these patients.”*
(S6, FG2)

Another subject highlighted the cost of the medication may still be very high even after the patient’s insurance is applied.


*“Yeah, I’ve seen- there are- so it’s normally like $3000 to $4000 depending on what kind of- well for the PrEP, specifically, it’s like roughly $2000 to $3000 for the cash price.”*
(P13, FG3)


*“However, I have been approached about like different programs which assist in getting programs to get it covered, so that’s kind of like the main issue that I see is that even when they have insurance, it will- sometimes, it will still be $100, and that may be too much for them to pay every month…”*
(S13, FG3)

2.Sub-theme 2: Stigma: “They just want the pills in an unmarked bottle”

This sub-theme presents the subject’s views on how stigma impacts patients taking PrEP. For example, the focus group discussion emphasized the impact stigma may have on a patient’s behavior. One subject suggested that receiving a medication bottle with the medication name might be considered invasive for some patients due to the stigma associated with the disease.


*“…Stigma surrounding HIV in general makes things challenging for patients. A lot of times patients don’t want to be identified so it’s hard when you call patients. They really just want the pills in an unmarked bottle. That’s one of the biggest things that we face issues with is just the stigma surrounding HIV still to this day that just puts a shade over everything we do with our HIV patients.”*
(S11, FG2) 

Another subject echoed the stigma impact on the patients and emphasized the limited information for the public to feel comfortable discussing the PrEP regimen with the pharmacist.


*“So, the approach to difficult conversations but also just the lack of knowledge surrounding HIV, the stigma associated with it, and how to identify patients at risk outside of just the fact of also knowing the medication regiments that are out there…So, kind of a robust answer, which is multi-factorial, but I think it’s a lot around the lack of knowledge on the provider’s side of pharmacy…”*
(S10, FG2)

Another subject explicitly links the medication, in this case, PrEP, to stigma.


*“I would say…stigma… you know that you’re coming to pick up this medication [PrEP] every month, you know, other people might know what they are taking and what it is for…”*
(S18, FG4)

The following quotation notes the significance of treating all patients equally and eliminating stigma to improve their healthcare outcomes. 


*“I think particularly patients with highly stigmatized conditions, it takes kind of going the extra mile, in a lot of cases, to show patients that you value them just as much as, you know, any other patient, and you don’t think of them any differently.”*
(S16, FG4)

3.Sub-Theme 3: Lack of patient knowledge of PrEP

This sub-theme expresses subjects’ opinions about the patient’s lack of knowledge about PrEP. In three FG discussions, the subjects echoed their main concern that patients lacked awareness of PrEP. In addition, there were some other discussions about the patient’s lack of understanding about the benefit of taking PrEP. For example, this quotation outlines a subject’s perception about PrEP awareness.


*“I think that there’s also a lack of information to the population. Some people don’t know that there is something called PrEP to protect themselves.”*
(S12, FG3)

Although the limited knowledge about PrEP represents a barrier for some patients, the subjects discussed possibilities on how to overcome this obstacle. Frequently, the subjects mentioned the usage of various brochures or social media as a means to disseminate information. For example, this excerpt demonstrates the importance of talking about these issues and working together to identify innovative solutions for this population.


*“I agree. Commercials, I mean, that’s the biggest way to get to a mass audience… But I think also, you know, people have to have a motivation to want to learn about these things, and, you know, with the stigma and cultural differences, those are huge barriers, and I don’t know how to get across those barriers…”*
(S12, FG3)


*“Yeah, like if there was a way that other users won’t know who is following or who is accessing it because that could be a really big barrier.”*
(S14, FG3)

The following quotation identifies a growing need for pharmacists to serve as a valuable resource for patients seeking care since they are a trusted and reliable member of the healthcare team.


*“I think practitioners have relationships with certain pharmacies. If the practitioner, once they see them, can guide them to the resources such as our pharmacy where we help patients get the assistance they needed. But I think if they’re not directed to the right resources that might be a barrier itself.”*
(S14, FG3)

### 3.2. Theme 2: Potential Solutions to Address Barriers to PrEP

The subjects provided viable solutions to address the obstacles to accessing PrEP that could improve patient outcomes.

The sub-theme 1: Solutions to address financial constraints

Although the FG discussions highlighted the cost of PrEP as the main barrier for patients, the subjects mentioned various ways practicing pharmacists could overcome these barriers. For example, several of the subjects offered examples of resources on how practicing pharmacists may address the high cost of PrEP.


*“There’s also co-pay assistance through the drug companies… There’s Advancing Access, which is a program that helps patients, especially if they are a lower-income patient, that they can get medication for free.”*
(S12, FG3)


*“I agree with [P12] and there’s also- the other resources…there’s foundations, the AIDS Foundations, and there’s always these links to multiple, multiple different types of co-payment programs.”*
(S14, FG3)

A few subjects practicing in a hospital setting highlighted how collaboration with other services can help to improve patient care and outcomes.


*“I work at the [Name] Hospital, and I work in the out-patient pharmacy sometimes. And so, often times when patients discharge and if they’re not insured, then we will work with case management to get their medications paid for by case management and social work or try to get them set up with someone who can do their medications for them.”*
(S4, FG1)

A number of subjects emphasized the importance of using certain medication programs that could alleviate the high cost of the monthly treatment.


*“…for some people, cost, this one is- especially in the Memphis areas, so many programs to assist people with the cost of their medication…”*
(S18, FG4)

The following quotation captures the significance of the discussion between the patient and the pharmacist and how the financial pressures might have made that patient non-adherent to PrEP.


*“… I had one patient who- I don’t know if he was afraid to bring it with us or if he just didn’t know, but I noticed when I was working at [Company Name], he was paying more than he really probably should be for his PrEP, and so I asked him if he had done one of the manufacturer coupons because it would have probably been free. And I think just kind of opening that door like, hey, I started this conversation, made a huge difference.”*
(S14, FG4)

2.Sub-theme 2: Solutions to address lack of patient knowledge

A number of the subjects provided solutions on how to tackle the general public’s lack of knowledge about PrEP. For example, several subjects recommended different avenues, including brochures, pamphlets, and QR codes, to market PrEP to the general public through a pharmacy. Furthermore, a few subjects noted the importance of mentioning to the patients that pharmacists have a role in addressing their questions about PrEP.


*“I think that would be good or maybe even something more subtle, like someone had mentioned a QR code earlier so maybe just like almost a little business card size. And that way they could take something and maybe it would be more discreet, especially for patients who are worried about stigma. And then they could go home, scan it, look at it on their phone in the privacy of their own home and not have this colorful pamphlet hanging around if that’s a concern for them.”*
(S7, FG2)

The following quotation demonstrates the importance of marketing PrEP through different pathways to enhance patient knowledge and respect their privacy.


*“*
*I think that there are several things here to kind of address. One being a pamphlet is very passive for some, but for others it’s very impactful. A QR code could be very passive for some, but very impactful. So thinking of how we kind of diversify our education strategies to meet the needs of the entire population that we’re talking about.”*
(S9, FG2)


*“I think that as a pharmacist, if we are able to go to these different places where we know that there is a higher population- you know, health fairs would be awesome. If we could somehow have a health fair, I think, going back to [previous]’s point, it would have to be aware of the culture of where we are in different parts of the cities… to see what pharmacists would be best suited.”*
(S14, FG3)

## 4. Discussion

Thematic analysis revealed two main themes: (1) Barriers to accessing PrEP; and (2) Potential solutions to address the barriers identified. These findings highlighted barriers and identified solutions to improve access to PrEP in Tennessee. For example, additional financial assistance programs and marketing programs targeting patients and providers are needed to enhance PrEP access.

One of our main themes reinforces the prominent issue that the cost of PrEP could hinder PrEP uptake, which is supported by previous literature on this topic [15,16]. Kay et al. have shown that the cost of PrEP is a significant obstacle to the utilization of PrEP across the U.S. and that insurance coverage—or a lack thereof—can influence the use of PrEP in populations where it is most warranted [15]. This extends to both the provider’s hesitancy to prescribe due to the cost and the patient’s inability to afford the medication [16]. There is an opportunity for pharmacists to play a major role by connecting the patient with the appropriate medication assistance program, thus enhancing uptake. Our findings align with prior literature and emphasize that the role of pharmacists is crucial in starting a conversation with a patient and connecting them to the appropriate resources.

For some of the participants, men who have sex with men (MSM), when users are adherent to PrEP in McKenney’s study, the authors attributed the low uptake of PrEP in the U.S. due to the financial burden [16]. This financial obstacle was corroborated by another study that surveyed clinicians and prescribers in various clinics across the South [17]. They found that prescribers were hesitant to prescribe PrEP due to the perception of PrEP costs and insurance coverage, especially outside of primary care clinics [17].

Many studies have also shown that healthcare providers have poor knowledge, comfort, and/or attitudes toward PrEP, and this may be related to gaps in PrEP education in healthcare training programs [18,19,20]. Pharmacists in this study not only acknowledged the need to enhance patient education and motivation toward PrEP, but they also understood the impact of their roles in understanding PrEP [21,22]. Therefore, our findings highlight the need for more longitudinal studies to explore the role of pharmacists in helping the patients identify evidence-based, affordable, and available options to mitigate the high cost of PrEP.

Although pharmacists in this study did not offer any solutions for addressing PrEP-related stigma, this is still important to note as this highlights the complexity of this social phenomenon. HIV stigma is devastating on familial, social, and economic levels, and it is very challenging to reduce as it is also difficult to define, measure, and accurately assess the impact [23]. PrEP is “stigmatized by association” because PrEP medications can also be used to treat HIV and people fear being associated with a disease process that is “socially discrediting” [24]. HIV stigma and, subsequently, PrEP stigma are very complicated topics that take a high level of commitment to address, and which may lead to pharmacists and other healthcare professionals feeling overwhelmed and mentally defeated. Similar to interventions to address HIV stigma, interventions to address PrEP stigma must be multifaceted and multilevel that address individuals, communities, and institutions in order to see improvements in PrEP access [23].

### Strength, Limitations, and Considerations

This study included a heterogeneous sample of Tennessee pharmacists practicing in different areas, representing a strength. Although the sample does not represent all Tennessee pharmacists, the study had a diverse representation from different areas of the state. These findings are not generalizable to all Tennessee pharmacists.

Using a quantitative design and the framework offered advantages in capturing the attitudes and perspectives of pharmacists practicing in different areas. Furthermore, qualitative research uncovered additional nuances in finding solutions to obstacles such as cost and stigma surrounding PrEP.

## 5. Conclusions

Pharmacists are optimally positioned to close the gap in PrEP care due to their expertise and accessibility. These findings highlight the barriers to accessing PrEP and provide various solutions to address unmet needs. Alternative options such as assistance programs and marketing programs targeting patients and providers are needed to enhance PrEP uptake.

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
