# Peer review of "Tennessee Pharmacists’ Opinions on Barriers and Facilitators to Initiate PrEP: A Qualitative Study"

_ijerph, 2022, doi:10.3390/ijerph19148431_

Round 1
Reviewer 1 Report
Overall:
My most substantive comment from the last submission – about the organization of the themes – was mostly addressed. However, after the suggested reorganization, there are still some quotes which seem to have been left to appear under the wrong theme/sub-theme. (See details of what I mean under “Results” below.) Additionally, some of my comments were not adequately addressed. More details are still needed in the Methods section (see below). Importantly, the entire manuscript still suffers from a very obvious lack of proofreading. There are problems with sentence clarity, grammar, and punctuation throughout. Please proofread your work! The organization of the Discussion section is still very problematic as well (see details below). Re-organization and re-writing of the Discussion section is the largest needed remaining revision.
Methods:
You did not address my comment from the last submission: “The third topic that was listed as part of the focus group guide seemed to be quite leading. However, this may just be because of a lack of adequate information presented about the content of the focus group guide. 'Recommendations for enhancing access to PrEP directly from the pharmacy' seems to inherently assume that all participants would already think it is a good idea for patients to be able to access PrEP directly from the pharmacy. What if some participants did not believe this? The way this topic is framed seems to suggest that there would be no room in the discussion for participants to voice potential concerns about this intervention strategy… However, this is only my assumption, as a list of 3 topic names is really not an adequate description of the focus group guide. It would be very important for the reader to have a fuller and clearer description of the topics in the guide.”
In particular, it is important to address the last sentence of this previous comment. More detail is needed about the focus group guide and its content.
More details are also needed on how coding discrepancies were addressed. You stated that “The team ensured a unified system of coding was obtained.” How?
Results:
There are still at least 4 sentences in results that are italicized but should not be (because italics were used to connote quotations only). Again, please proofread your work.
The quotes where the participants are suggesting using social media and commercials to increase knowledge were put into the barriers theme… they should be in the “solutions to address lack of knowledge” theme/subtheme. The participants seem to have generated the ideas of addressing the problem with commercials and social media so… that isn’t the barrier. That’s the solution. And the specific quotes to which I’m referring don’t go into detail about the barrier itself at all anyway. They’re entirely solution-focused. At least 2 or 3 such quotes should be moved accordingly.
Discussion:
The first sentence in the paragraph starting on line 338 is nonsensical, both grammatically and in terms of content and its presentation. Again, please proofread your work.
The organization of the Discussion is still very, very poor. It is perhaps more disorganized than the last submission. The Discussion should always open with a summary of the key findings of your own study (discussed vis a vis the extant literature). In your last submission, you had a summary, which you seem to have now simply removed since it presented themes that contradicted those presented in your Results section in the last draft. You can’t just omit the summary, however. You need to include one that matches the themes you have now finalized. Your first paragraph in this draft of the Discussion makes recommendations for practice and policy (which should appear near the end of the Discussion) before you have even stated your findings and their meaning. This doesn’t work. Your last paragraph (before the Limitations subsection) – i.e., the paragraph starting on line 338 - in this draft of the Discussion is attempting to link your findings to the extant literature on financial burden, but does not do so clearly (because your own findings are not actually mentioned at all). And again, this link between your own findings and this literature needs to appear at the beginning of the Discussion. The new addition of the paragraph focused on stigma is excellent. But again, the information both before and after that new paragraph needs to be much better organized and well-written (i.e., written with clearer meaning and certainly with a clearer summary of - and mentions of throughout - your own findings).
Author Response
My most substantive comment from the last submission – about the organization of the themes – was mostly addressed. However, after the suggested reorganization, there are still some quotes which seem to have been left to appear under the wrong theme/sub-theme. (See details of what I mean under “Results” below.) Additionally, some of my comments were not adequately addressed. More details are still needed in the Methods section (see below). Importantly, the entire manuscript still suffers from a very obvious lack of proofreading. There are problems with sentence clarity, grammar, and punctuation throughout. Please proofread your work! The organization of the Discussion section is still very problematic as well (see details below). Re-organization and re-writing of the Discussion section is the largest needed remaining revision.
Methods:
You did not address my comment from the last submission: “The third topic that was listed as part of the focus group guide seemed to be quite leading. However, this may just be because of a lack of adequate information presented about the content of the focus group guide. 'Recommendations for enhancing access to PrEP directly from the pharmacy' seems to inherently assume that all participants would already think it is a good idea for patients to be able to access PrEP directly from the pharmacy. What if some participants did not believe this? The way this topic is framed seems to suggest that there would be no room in the discussion for participants to voice potential concerns about this intervention strategy… However, this is only my assumption, as a list of 3 topic names is really not an adequate description of the focus group guide. It would be very important for the reader to have a fuller and clearer description of the topics in the guide.”
In particular, it is important to address the last sentence of this previous comment. More detail is needed about the focus group guide and its content.
Response: We value your suggestions. We made changes in the methods, and we provided a sample of our FG questions.
More details are also needed on how coding discrepancies were addressed. You stated that “The team ensured a unified system of coding was obtained.” How?
Response: Thank you for your suggestion. We have added a brief description of the process.
Results:
There are still at least 4 sentences in results that are italicized but should not be (because italics were used to connote quotations only). Again, please proofread your work.
Response: Thank you for your observation. We have amended the text.
The quotes where the participants are suggesting using social media and commercials to increase knowledge were put into the barriers theme… they should be in the “solutions to address lack of knowledge” theme/subtheme. The participants seem to have generated the ideas of addressing the problem with commercials and social media so… that isn’t the barrier. That’s the solution. And the specific quotes to which I’m referring don’t go into detail about the barrier itself at all anyway. They’re entirely solution-focused. At least 2 or 3 such quotes should be moved accordingly.
Response: Thank you for your suggestion. We have moved those quotations into the results section hat focuses on the second major theme.
Discussion:
The first sentence in the paragraph starting on line 338 is nonsensical, both grammatically and in terms of content and its presentation. Again, please proofread your work.
Response: Thank you for your recommendation. We have amended the text.
The organization of the Discussion is still very, very poor. It is perhaps more disorganized than the last submission. The Discussion should always open with a summary of the key findings of your own study (discussed vis a vis the extant literature). In your last submission, you had a summary, which you seem to have now simply removed since it presented themes that contradicted those presented in your Results section in the last draft. You can’t just omit the summary, however. You need to include one that matches the themes you have now finalized. Your first paragraph in this draft of the Discussion makes recommendations for practice and policy (which should appear near the end of the Discussion) before you have even stated your findings and their meaning. This doesn’t work. Your last paragraph (before the Limitations subsection) – i.e., the paragraph starting on line 338 - in this draft of the Discussion is attempting to link your findings to the extant literature on financial burden, but does not do so clearly (because your own findings are not actually mentioned at all). And again, this link between your own findings and this literature needs to appear at the beginning of the Discussion. The new addition of the paragraph focused on stigma is excellent. But again, the information both before and after that new paragraph needs to be much better organized and well-written (i.e., written with clearer meaning and certainly with a clearer summary of - and mentions of throughout - your own findings).
Response: Thank you for your suggestions. We reorganized the discussion and made the suggested changes.
Reviewer 2 Report
I thank the authors for addressing my previous commemnnts.
Author Response
I thank the authors for addressing my previous commemnnts.
Response: You are welcome.
This manuscript is a resubmission of an earlier submission. The following is a list of the peer review reports and author responses from that submission.
Round 1
Reviewer 1 Report
Thank you for inviting me to read and comment on this manuscript that describes a qualitive study to assess and characterise Tennessee pharmacists’ perceptions about the barriers and facilitators to widespread usage of PrEP. The idea for this work is interesting and the reported study, which is well-designed and nicely conducted it would be of interest for the readers of the journal. It cocludes that pharmacy-led PrEP programs may overcome existing barriers to widespread PrEP utilization, which is especially needed in Tennessee.
Reviewer 2 Report
This manuscript presents a qualitative study of pharmacists’ perceptions of barriers and facilitators to accessing PrEP. The Background section does a compelling job of explaining that this study addresses a gap in the literature and that pharmacists represent a key untapped mechanism of intervention for facilitating access to PrEP. I believe this paper could make an important contribution to the literature.
Unfortunately, however, after the Background section, the paper is poorly organized not clearly written in some areas. (And it is sloppily written in other areas… e.g. the first paragraph of Discussion and the paragraph in Conclusion are almost identical to each other; within the Discussion section there is a sentence that appears twice in a row with one or two words changed in its second appearance; the authors attempted use italics to indicate a participant quotation but in at least 4 places in the Results section they have italicized sentences that are not quotations, making it hard for the reader to discern what is happening.)
The biggest and most problematic example of the poor organization of the paper is that the Abstract and Results section present 3 themes that were found from the qualitative analysis… but then the Discussion and Conclusions sections state that 3 themes were found and describe them as being 3 DIFFERENT themes than the ones that were presented in the Abstract and Results! This reads as though two different people who did not actually agree on what the themes were that emerged from the analyses wrote these different sections of the paper, without reading each other’s work afterwards… (I acknowledge that that may not at all be what actually happened… that is just how it appears from a reader’s perspective.) Any attempt to resubmit this paper to this or another journal simply must entail presentation of the same findings consistently throughout all sections of the paper.
Within the Results section, the way that the themes presented there are organized does not really seem to make much logical sense. Specifically, if Theme 1 is “Obstacles that hinder PrEP access and uptake,” then how is “financial constraints” not a subtheme (and not really considered by the authors as part of Theme 1 at all)? The authors explicitly say (as part of Theme 2) that the participants think that financial constraints are the biggest barrier… so using logic, that concept of financial obstacles certainly must then be the most important subtheme under Theme 1: “obstacles that hinder PrEP access and uptake,” right? But the topic of financial barriers is somehow mostly ignored by the authors under Theme 1, where it clearly, logically should be included. The only subtheme named under Theme 1 is “What is PrEP?”… by which the authors are trying to refer to “lack of knowledge of PrEP” (which would be a much clearer and more appropriate name)…. But the authors’ presentation and discussion of the data for Theme 1 seem to very clearly suggest that stigma is another major subtheme under Theme 1, as – again – is financial constraint. So if you are going to name any subthemes of “obstacles that hinder PrEP access and uptake,” then you should create subthemes for all different obstacles that are repeatedly and emphatically emphasized (to the same degree as lack of knowledge) by the participants. It seems very, very clear from the quotes provided that the participants were repeatedly describing financial barriers and stigma as major barriers.
It’s also not clear to the reader why “Solutions to financial constraints” became its own theme, when the many potential solutions the participants offered to the barrier of lack of knowledge did not become its own theme or subtheme…
From an organizational standpoint, it seems like it would be logical to have Theme 1: barriers to accessing PrEP, with subthemes financial constraints, stigma, and lack of knowledge. Then Theme 2: potential solutions to address barriers to accessing PrEP, with subthemes solutions to address financial constraints and solutions to address lack of knowledge… If the data and analysis contraindicated this organization of themes, it isn’t clear to the reader, and that may be in part because the pertinent details about the specific analytic approach that was used are lacking from the Methods section.
In fact, there are many details that should be in the Methods section that are either missing altogether or are unclearly written such that the reader cannot discern what was done. Again, the most noteworthy thing, in my opinion, is the specific analytic strategy used. “Thematic analysis” is not actually a specific qualitative analytic approach. All qualitative analyses which arrive at themes (which is the vast majority of them) are technically thematic analyses. You did mention that the analysis was inductive, which is an extraordinarily important thing to specify… but what kind of inductive approach did you use, specifically? Grounded theory? Constant comparative analysis? Etc.
The third topic that was listed as part of the focus group guide seemed to be quite leading. However, this may just be because of a lack of adequate information presented about the content of the focus group guide. “Recommendations for enhancing access to PrEP directly from the pharmacy” seems to inherently assume that all participants would already think it is a good idea for patients to be able to access PrEP directly from the pharmacy. What if some participants did not believe this? The way this topic is framed seems to suggest that there would be no room in the discussion for participants to voice potential concerns about this intervention strategy… However, this is only my assumption, as a list of 3 topic names is really not an adequate description of the focus group guide. It would be very important for the reader to have a fuller and clearer description of the topics in the guide.
The last sentence in the Methods section mentions that two additional researchers were brought in… and it seems that their purpose was to validate the findings of the analysis… however this important piece of information is not clearly written. It is not clear from this sentence what the two additional researchers did. It says they “read the quotes and reflections of the words and the intended meanings of the FG.” First of all… how can they “read the intended meanings of the focus group”??? Did you mean that they tried to infer the intended meanings? You didn’t actually say what they DID after they read the material. This is important and needs clarified.
The Limitations section is poorly written and some of the sentences therein do not make sense. It uses the word “quantitative” mistakenly instead of “qualitative.” But that is trivial. More importantly, sentences within this section seem to contradict each other. On the one hand, the authors have made the point that one strength of their study was that they included pharmacists working in multiple types of settings, but on the other hand, they then say that future research should recruit only pharmacists in community settings? This doesn’t make sense… but the sentences are unclearly written, such that I’m not even entirely sure that those are the two points the authors were attempting to make. The last sentence of the Limitations section then refers to “this population”…. when there was absolutely no mention whatsoever of any population in the entire Limitations section. So… what population? This writing is disjointed and impossible to follow.
This is an important research topic and probably a good study (though more details are needed in Methods to be sure), and I would love to see it published, but unfortunately, the execution of this manuscript at least (and probably also the qualitative analysis, since the presentation of themes was both illogical and contradictory between sections of the paper) need to be dramatically improved before publication should occur.
Reviewer 3 Report
This is an interesting article on a relevant topic. I have some minor comments below that I feel could improve the manuscript:
- Line 12: A verb is missing here.
- On lines 39-40 the authors state that: “Furthermore, of the 1 million Americans who could benefit from PrEP, only 25% are using this prevention 4” However, on lines 46-47 the data quoted are repetitive but somewhat different: “75% of the 1.2 million people requiring PrEP do not receive it, particularly among minority populations.8”
- Lines 51 and 54: Why is the reference name also cited in parentheses?
- Line 116: The number of subjects who declined to participate would be best presented in a flowchart depicting how many invitations were sent, how many invitations were accepted and how many participants dropped out after listening to the ICF, or during or after the interview.
- Lines 160-161, 238-239 should probably not be italicized.
- The Discussion is generally well structured, but there is information repeating the data already presented in the information, i.e., that pharmacists are optimally positioned to close the gap in PrEP care, the “purview paradox”, etc. I would recommend that the authors re-read the Introduction and Discussion together to be able to streamline the information presented while avoiding repetitions.
- Are there any successful implementation examples/pilot projects that the authors could reference?
- PrEP should be discussed in the context of the 90-90-90 targets in Tennessee and how close is the state to the first “90”.
- I see that there was IRB approval for this study. Was there also an ethics approval obtained? Along with an approval for oral informed consent?